# Estimation of Winter Wheat Residue Coverage Using Optical and SAR Remote Sensing Images

**Wenting Cai** [1,2,3], **Shuhe Zhao** [1,2,3,*], **Yamei Wang** [1,2,3], **Fanchen Peng** [1,2,3], **Joon Heo** [4] **and Zheng Duan** [5]

1   School of Geography and Ocean Science, Nanjing University, Nanjing 210023, China; MG1727099@smail.nju.edu.cn (W.C.); MG1827104@smail.nju.edu.cn (Y.W.); MG1827068@smail.nju.edu.cn (F.P.)

2   Jiangsu Center for Collaborative Innovation in Geographical Information Resource Development and Application, Nanjing 210023, China

3   Collaborative Innovation Center of South China Sea Studies, Nanjing University, Nanjing 210023, China

4   Department of Civil and Environmental Engineering, Yonsei University, Seoul 03722, Korea; jheo@yonsei.ac.kr

5   Lancaster Environment Centre, Lancaster University, Lancaster LA1 4YQ, UK; z.duan@lancaster.ac.uk

*   Correspondence: zhaosh@nju.edu.cn

**Abstract:** As an important part of the farmland ecosystem, crop residues provide a barrier against water erosion, and improve soil quality. Timely and accurate estimation of crop residue coverage (CRC) on a regional scale is essential for understanding the condition of ecosystems and the interactions with the surrounding environment. Satellite remote sensing is an effective way of regional CRC estimation. Both optical remote sensing and microwave remote sensing are common means of CRC estimation. However, CRC estimation based on optical imagery has the shortcomings of signal saturation in high coverage areas and susceptibility to weather conditions, while CRC estimation using microwave imagery is easily influenced by soil moisture and crop types. Synergistic use of optical and microwave remote sensing information may have the potential to improve estimation accuracy. Therefore, the objectives of this study were to: (i) Analyze the correlation between field measured CRC and satellite derived variables based on Sentinel-1 and Sentinel-2, (ii) investigate the relationship of CRC with new indices (OCRI-RPs) which combine optical crop residues indices (OCRIs) and radar parameters (RPs), and (iii) to estimate CRC in Yucheng County based on OCRI-RPs by optimal subset regression. The correlations between field measured CRC and satellite derived variables were evaluated by coefficient of determination ($R^2$) and root mean square error (RMSE). The results showed that the normalized difference tillage index (NDTI) and radar indices 2 (RI2) had relatively higher correlations with field measured CRC in OCRIs and RPs ($R^2$ = 0.570, RMSE = 6.560% and $R^2$ = 0.430, RMSE = 7.052%, respectively). Combining OCRIs with RPs by multiplying each OCRI with each RP could significantly improve the ability of indices to estimate CRC, as NDTI × RI2 had the highest $R^2$ value of 0.738 and lowest RMSE value of 5.140%. The optimal model for CRC estimation by optimal subset regression was constructed by NDI71 × $\sigma_{VV}^0$ and NDTI × $\sigma_{VH}^0$, with a $R^2$ value of 0.770 and a RMSE value of 4.846%, which had a great improvement when compared with the best results in OCRIs and RPs. The results demonstrated that the combination of optical remote sensing information and microwave remote sensing information could improve the accuracy of CRC estimation.

**Keywords:** crop residues coverage; optical crop residue indices; radar parameters; Sentinel-1; Sentinel-2; optimal subset regression; winter wheat

## 1. Introduction

Crop residue is defined as stalks, leaves and any other plant litter of crops such as maize, wheat, soybean, rice, etc., which accumulates on the surface of farmland after harvest [1,2]. Crop residues play an important role in the conservation of tillage systems, which include reducing water-based and wind-based soil erosion, increasing soil organic matter content, fixing $CO_2$ in the soil and improving soil quality [3–5]. It also has positive influences on water infiltration and evaporation by improving soil structure and reducing daily variation of soil temperature [6,7]. Consequently, good residue management practices contribute significantly to an increase of crop yields [8–10]. From the view point of air quality protection, remaining crop residues in farmland can greatly reduce the emission of toxic gasses such as CO, $SO_2$ and $NH_3$, while burning straws severely pollute the air [11,12]. Moreover, crop models need crop residue coverage (CRC) as an input parameter to simulate the impact of management practices on crop production. Therefore, it is of great importance to estimate CRC in crop planting areas and have a knowledge of the population of conservation tillage on a regional scale.

Remote sensing is a useful tool to estimate CRC efficiently on a regional scale in a timely manner [13–16]. Many attempts have been done to explore the relationship between CRC and satellite derived variables. Among them, regression-based spectral models were the most commonly used. They are based on the crop residue indices derived from optical remote sensing images, which included Normalize Difference Tillage Index (NDTI) [17], Normalized Difference Senescent Vegetation Index (NDSVI) [18], Normalized Difference Residue Index (NDRI) [19], Normalized Difference Index 5 (NDI5) [20], Normalized Difference Index 7 (NDI7) [20], and so on. Previous studies have proved that NDTI performed best in most cases [21], while NDSVI was more suitable in humid surroundings which avoided using the infrared bands (2080–2350nm) of Landsat TM. In the meanwhile, NDRI was proposed for the situation where green vegetables have appeared in the field. Thus, crop residue indices based on multispectral images are an integral part of the CRC estimation. However, as other indices used in the field of forest biomass estimation, and other quantitative remote sensing, crop residue indices have a poor performance in high coverage areas, which was named as the "saturation" phenomenon [22]. It largely underestimated CRC in the range of 80–100% and influenced the overall accuracy of CRC estimation [16,22].

Compared with optical remote sensing information, microwave remote sensing has the advantage of not being sensitive to weather conditions. Previous studies have made good progress in CRC estimation based on microwave remote sensing information [23–27]. The ground experiments have found that the backscattering coefficient of microwave scatterometer in a specific polarization direction has a good correlation with CRC of a specific crop type, which made it possible to estimate CRC by Synthetic Aperture Radar (SAR) data [23,24]. McNairn et al. [25] also demonstrated the effectiveness of spaceborne SAR data in CRC estimation by using Spaceborne Imaging Radar-C data, and the parameters of cross-polarized linear backscatter and co-polarized circular backscatter were proved to be suitable for CRC estimation. Therefore, SAR imagery has great potential to estimate CRC. However, McNairn et al. also highlighted that the contribution of crop residues to radar response was only 40% [25], which made it difficult to estimate crop residue cover using microwave responses alone. The estimation accuracy was easily influenced by residue type and condition, incidence angles, surface roughness and so on [24,27,28]. All of these problems limited the use of SAR image in CRC estimation.

The estimation of CRC using optical or microwave data alone often has their own shortcomings. Studies have shown that the method combining optical and microwave remote sensing information had a better result than methods only using one type of remote sensing data in the fields of forest biomass estimation, leaf area index (LAI) estimation and other fields [29–35]. Jin et al. [29] estimated LAI and biomass of winter wheat with combined optical spectral vegetation indices and radar polarimetric parameters, which provided a guideline for the estimations of LAI and biomass using multisource remote sensing data. Huang et al. [30] proved that the synergistic use of Advanced Spaceborne Thermal Emission and Reflection Radiometer (ASTER) and European Remote-Sensing Satellite-2 (ERS-2) SAR could estimate above ground biomass more accurately than using one type of remote

sensing information in Xixi National Wetland Park, China. Cutler et al. [31] combined the textural information of JERS-1 SAR image with Landsat TM to estimate forest biomass in three regions and the estimation accuracy also had significantly improvements. Synergistic use of optical imagery and SAR imagery can significantly improve the predicted accuracy of LAI and biomass. Therefore, exploring the effectiveness of the combined method for CRC estimation is of crucial importance.

Sentinel-1 and Sentinel-2 were both launched by the European space agency from 2014 to 2017. They have opened up opportunities for techniques which are aimed at improving the accuracy of CRC estimation. Sentinel-1 is a C-band synthetic aperture radar mission that provides data covering most parts of the world. Compared with other radar satellites in orbit, Sentinel-1 has a higher spectral resolution of $1dB/3\sigma$, which makes it more suitable for quantitative estimation, and the wavelength of Sentinel-1 is relatively short. That makes it more likely to be effective for tillage assessment [28]. Sentinel-2 is a multispectral satellite mission that sets up 13 bands from coastal wavelengths to short-wave infrared wavelength with spatial resolutions of 10, 20, or 60 m. Sentinel-2 can provide much more spectral information than Landsat OLI, which has made it more advantageous in quantitative estimation. Both Sentinel-1 and Sentinel-2 missions are in constellation with two twin satellites. Hence, the time resolution of Sentinel satellites can reach up to 6 days. The balance between spatial resolution and time resolution make the Sentinel-1 and Sentinel-2 more suitable for agricultural monitoring. Therefore, using Sentinel-1 and Sentinel-2 to explore the effectiveness of the method combining optical and microwave data for CRC estimation may have more meaning in the future applications.

Winter wheat is a main crop in the North China Plains, which is one of the most important food production regions in China. Estimating winter wheat residue coverage is important for agricultural production and environment protection. Therefore, the objectives of this study were to (i) investigate the potential of Sentinel-2 and Sentinel-2 for CRC estimation using several optical crop residue indices (OCRIs) and five radar parameters (RPs), (ii) investigate the relationship of CRC with new indices which combined OCRIs with RPs, and (iii) build up an optimal model for CRC estimation using OCRI-RPs by optimal subset regression. The findings of this study were expected to provide a guideline for accurately estimating CRC based on optical information and SAR information on a landscape scale.

## 2. Materials and Methods

### 2.1. Study Area

This study was conducted in the farmland of Yucheng County, which belongs to the Shandong Province, China, and located in the center of the Huanghuaihai plains (Figure 1). Yucheng has a large area of farmland, which mainly grows winter wheat, maize, soybean, cotton and vegetables. The dominant crops are wheat and maize. The planting pattern is winter wheat and summer maize rotation. Winter wheat is sown in early October and harvested in the next year June. After the harvest, corn is planted on the farm. Yucheng is characterized with a warm, temperate, semi-humid, monsoon climate. The average annual temperature is 13.1°C, and the average annual precipitation is 600 nm. The frost-free period lasts for 200 days. The majority of the soil type is silt and light flux loam, which comes from the Yellow River. Yucheng is one of the biggest grain production bases in China. Therefore, estimating CRC in Yucheng County and understanding the implementation of conservation tillage practices is meaningful.

### 2.2. Field Measurement Data

The field measurement data was collected over two years. One was from 13 June 2017 to 18 June 2017 and another was from 10 June 2018 to 20 June 2018. In 2017, eight samples were set up in the winter wheat cultivation area. While in 2018, 30 samples were set up in the same study area. In the 30 samples collected in 2018, eight samples were too close to the road, which were not suitable for the calculation of spectral and textural information. Therefore, this study did not used these eight samples. In conclusion, a total of 30 samples were used for CRC estimation (Figure 1). The size of each samples

was 30 m × 30 m. In each sample, the information of 5 sample plots were collected to represent the sample. Each sample plot was 1 m$^2$ and distributed as Figure 2 shows. In the field measurement, iron wire frames (1 m × 1 m) were put on sample plots and photos were taken by camera to collect the CRC information of samples. Previous studies have proved that using cameras to collect CRC information (photo method) had a higher accuracy than the line transect method [16]. GPS was used to measure latitude and longitude information of each sample so that the field measurement location could be collocated with remote sensing images.

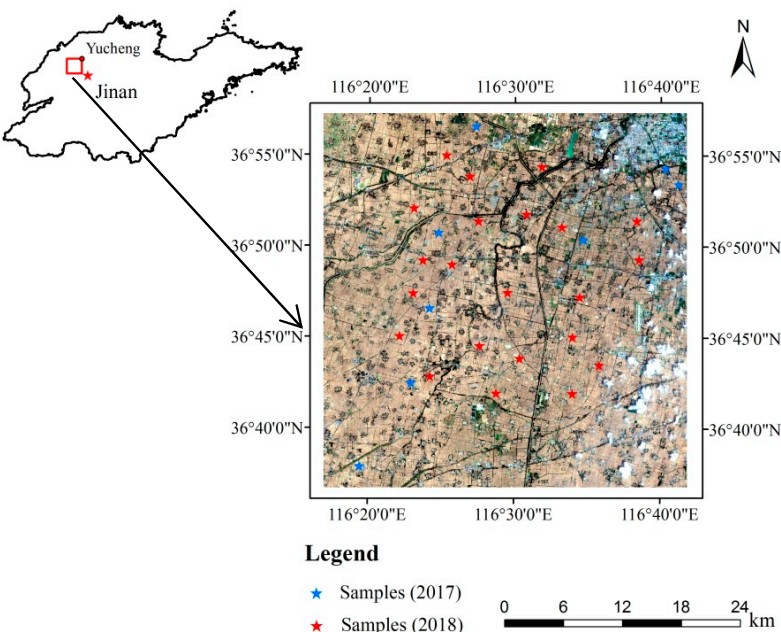

**Figure 1.** Study site, Yucheng County, Shangdong Province, China.

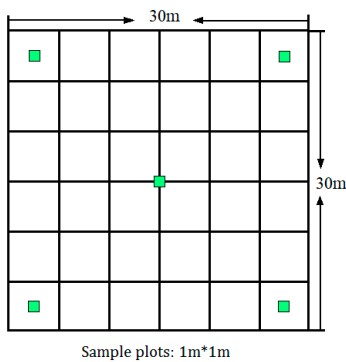

**Figure 2.** Distribution of sample plots in one Sample.

The CRC of sample plots could be obtained by distinguishing green vegetable, soil and crop residues in the photos. First, the photo needed to be clipped and only the part within the iron wire frame could remain. Second, eCognition software (provided by Definiens Imaging Company) was used to segment the photos. The parameter of segmentation scale was set to 20. In this scale, soil and crop residues can be distinguished well. Najafi et al. found that the brightness of image has a good correlation with CRC [36]. Hence, brightness was used as condition to distinguish soil and crop residues. After repeated experiments, this study used 50 as threshold value to identify soil objects from the photograph. Meanwhile, commission objects and omission objects were corrected by visual interpretation. Because green vegetables were seldom distributed in the photo, they were extracted by visual interpretation. Finally, the CRC of photo could be obtained by calculating the pixel number of crop residues. Averaging the CRC of five photos in one sample gave the CRC of sample. Figure 3

shows the local of one photo and the soil extraction results. It can be easily found that the soil extraction results were close to the truth.

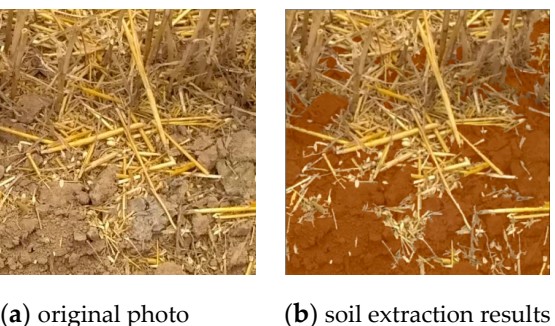

(**a**) original photo          (**b**) soil extraction results

**Figure 3.** Original photo and the processing results.

## *2.3. Satellite Data and Pre-Processing*

### 2.3.1. Sentinel-1 Imagery

In this study, two C-band Level-1 Ground Range Detected (GRD) Sentinel-1 SAR images were used to estimate CRC in Yucheng County. One image was acquired on 13 June 2017 and another was 14 June 2018. Both SAR images were used the Interferometric Wide (IW) swath mode with a spatial resolution of 5 m × 20 m and a swath of 250 km. The polarization directions were VV (Vertical Transmit/Vertical Receive—like polarization) and VH (Vertical Transmit/Horizontal Receive—cross polarization). The Sentinel-1 GRD data downloaded from the website of the European Space Agency were ground distance detected and projection converted. Additionally, four pre-processing steps were conducted by SNAP software, which included radiation calibration, noise filtering, geometric correction and resampling. At last, the normalized backscattering coefficient images in VV and VH polarization directions were obtained with a spatial resolution of 10 m. For the difference of two images in incidence angles were small (38.89° and 39.30°, respectively), this study ignored the estimation error from two SAR images.

### 2.3.2. Sentinel-2 Imagery

Two Sentinel-2 Level-1C images acquired on June 14, 2017 and June 14, 2018 were used in this study to explore the relationship between CRC and optical satellite derived variables. Also, two Sentinel-2 Level-1C images acquired on 10 May 2018 and 24 June 2018 were used to extract winter wheat cultivation area. Sentinel-2 images were downloaded from the website of the European Space Agency. They were geometrically corrected. Later, Sen2Cor was used to atmospherically correct the Sentinel-2 image, and SNAP software was used to resample the spatial resolution of bands to 10 m by nearest distance method.

## *2.4. Methods*

### 2.4.1. Calculations of Satellite Derived Variables

Previous studies have shown that NDI7, NDTI and NDRI, which are based on the infrared bands in 2100–2280 nm, performed better than NDI5 and NDSVI, which are based on the infrared bands in 1565–1655 nm in Yucheng County, when using Sentinel-2 data to quantify CRC [21]. Thus, this paper calculated these three indices to estimate CRC. Considering that Sentinal-2 has four additional bands in the 700–900 nm range, this paper replaced the near-infrared bands in the formula of NDI7 with the fifth band (b5, 694–714 nm), sixth band (b6, 730–750 nm), seventh band (b7, 768–797 nm) and 8A band (b8A, 848–882 nm) of Sentinel-2 to build four new crop residue indices, which were named as NDI71,

NDI72, NDI73 and NDI74 respectively. The formulas of selected crop residue indices are shown in Table 1.

**Table 1.** The selected optical crop residue indices used in this study.

| Vegetation Index | Abbreviation | Formula | Reference |
|---|---|---|---|
| Normalized Difference Residue Index | NDRI | (b4 − b12)/(b4 + b12) | [19] |
| Normalized Difference Index 7 | NDI7 | (b8 − b12)/(b8 + b12) | [20] |
| Normalized Difference Tillage Index | NDTI | (b11 − b12)/(b11 + b12) | [17] |
| Normalized Difference Index 71 | NDI71 | (b5 − b12)/(b5 + b12) | This paper |
| Normalized Difference Index 72 | NDI72 | (b6 − b12)/(b6 + b12) | This paper |
| Normalized Difference Index 73 | NDI73 | (b7 − b12)/(b7 + b12) | This paper |
| Normalized Difference Index 74 | NDI74 | (b8A − b12)/(b8A + b12) | This paper |

For SAR images, six radar parameters were used to explore the relationship between field measures CRC and radar images. The first two RPs were normalized backscattering coefficients in VV, and VH polarization directions ($\sigma_{VV}^0$, $\sigma_{VH}^0$). Beside them, two radar indices (RI) built by $\sigma_{VV}^0$ and $\sigma_{VH}^0$ were used. The formulas are shown in Equations (1) and (2). The mean variable of gray level co-occurrence matrix (GLCM) is a commonly used variable in quantitative estimation [37]. Thus, this study also considered the mean variables of $\sigma_{VV}^0$ and $\sigma_{VH}^0$ (VV_ME, VH_ME) to estimate CRC:

$$RI1 = \sigma_{VV}^0 / \sigma_{VH}^0 \tag{1}$$

$$RI2 = \left(\sigma_{VV}^0 - \sigma_{VH}^0\right) / \left(\sigma_{VV}^0 + \sigma_{VH}^0\right) \tag{2}$$

where $\sigma_{VV}^0$ and $\sigma_{VH}^0$ are the backscattering coefficients in VV, and VH polarization directions, respectively.

### 2.4.2. Optimal Subset Regression Method

Optimal subset is a method that selects all combinations of independent variables as subset to estimate possible dependent variables and then chooses the best one. For a model with *n* independent variables, it can produce 2n − 1 subsets. There are four principles for selecting the best subset: (1) coefficient of determination ($R^2$); (2) Bayes Information Criterion (BIC); (3) Akaike's Information Criterion (AIC); and (4) model validation accuracy. The subset is better if the $R^2$ is larger and BIC, AIC, and model validation accuracy are smaller. Making a comprehensive assessment of the four indicators a method to obtain the best subset.

If independent variables in a model have the problem of colinearity, it will affect the prediction ability of the model. This study used variance inflation factor (VIF) to estimate colinearity between independent variables [38]. It can be calculated by the following formula:

$$VIF = \frac{1}{1 - R_i^2} \tag{3}$$

where $R_i^2$ is the coefficient of determination, when the *i*th independent variable is set as the dependent variable, and other independent variables as independent variables to regress. It is generally believed that serious colinearity exists when VIF is more than 10. The process of optimal subset regression is realized in the R software (leaps package: https://cran.r-project.org/web/packages/available_packages_by_name.html).

### 2.4.3. Statistical Analysis

The coefficient of determination ($R^2$) was used in this study to measure the correlation between field measured CRC and satellite derived variables. The formula of $R^2$ is shown in Equation (4).

The satellite derived variable with higher $R^2$ value was considered as the most suitable variables to estimate CRC:

$$R^2 = \frac{\sum_{i=1}^{n}(x_i - x')(y_i - y')}{\sqrt{\sum_{i=1}^{n}(x_i - x')^2 \cdot \sum_{i=1}^{n}(y_i - y')^2}} \tag{4}$$

where $n$ is the total number of field measured samples, $x_i$ is the estimated CRC for $i$th sample, $y_i$ is the field measured CRC for $i$th samples, while $x'$ and $y'$ are the mean values of them.

Leave-one-out cross validation (LOOCV) was used to evaluate the accuracy of models in predicting CRC in the test set. It put every sample as test set in turn to evaluate the accuracy of model made by other samples, and calculated the root mean square error (RMSE) of each test set. The accuracy of the model depends on the value of RMSE in LOOCV. The accuracy is higher when RMSE is lower. The formula of RMSE is shown in Equation (5):

$$RMSE = \sqrt{\frac{1}{n}\sum_{i=1}^{n}(x_i - y_i)^2} \tag{5}$$

where $n$ is the total number of field measured samples, $x_i$ is the eimated CRC for ith sample, $y_i$ is the field measured CRC for ith samples, $x'$ and $y'$ are the mean values of them.

### 2.4.4. Extraction of Winter Wheat Cultivation Area

Two Sentinel-2 images were used in this study to extract winter wheat cultivation area. Previous studies have shown that green vegetables have a high value of Normalized Difference Vegetable Index (NDVI), which would decline significantly when vegetables die away [39,40]. The NDVI of winter wheat in Yucheng County generally reaches a peak in early May and decline to the lowest value in late June [41]. In view of these, winter wheat cultivation area was obtained by calculating the difference of NDVI in two Sentinel-2 images, which were acquired on 10 May 2018 and 24 June 2018, and 0.4 was set as the threshold value to distinguish winter wheat cultivation areas from other farmlands. The extracted results well avoided the confusion between forest and other green vegetation.

## 3. Results

### 3.1. Correlation between Optical Crop Residue Indices and Field Measured CRC

Linear regression analysis was applied to build the relationship between seven selected optical crop residue indices (OCRIs) and field measured CRC. The regression equations and coefficients of determination are shown in Table 2. The table shows that the correlations between OCRIs and CRC were significant. NDTI and NDI73 had the highest and lowest $R^2$ values of 0.570 and 0.157, respectively. The indices with $R^2$ from high to low were NDTI, NDI71, NDRI, NDI7, NDI72, NDI74 and NDI73. The corresponding $R^2$ values were 0.570, 0.472, 0.462, 0.425, 0.285, 0.238 and 0.157, respectively. Of the $R^2$ value, one was above 0.50, three were between 0.50 and 0.40 and three were below 0.40. The scatterplots between CRC and some OCRIs are shown in Figure 4.

RMSE was used to compare the ability of indices in predicting CRC values. It easily found that NDTI had the lowest RMSE of 6.560%, while NDI73 had the highest RMSE value of 9.155%. The orders of indices from lowest to highest with respect to RMSE were similar to the order of $R^2$ value from highest to lowest. The RMSE values of NDTI, NDI71, NDRI, NDI7, NDI72, NDI74 and NDI73 were 6.560%, 7.317%, 7.430%, 7.597%, 8.498%, 8.714% and 9.155%, respectively. In conclusion, most OCRIs had the ability to estimate CRC. Among them, NDTI performed best.

**Table 2.** The regression relationships between CRC and optical crop residue indices.

| Crop Residue Indices | Regression Equations | $R^2$ | RMSE (%) |
|---|---|---|---|
| NDRI | Y = 1.391x + 1.064 | 0.462 ** | 7.430 |
| NDI7 | Y = 1.299x + 0.820 | 0.425 ** | 7.597 |
| NDTI | Y = 4.000x + 0.360 | 0.570 ** | 6.560 |
| NDI71 | Y = 1.744x + 1.015 | 0.472 ** | 7.317 |
| NDI72 | Y = 1.10x + 0.864 | 0.285 ** | 8.498 |
| NDI73 | Y = 0.769x + 0.796 | 0.157 * | 9.155 |
| NDI74 | Y = 0.935x + 0.770 | 0.238 ** | 8.714 |

**Note**: CRC means the field measured crop residue coverage; ** means model significant at the 0.01 probability level; * means model significant at the 0.05 probability level. RMSE means the root mean square error of leave-one-out cross validation.

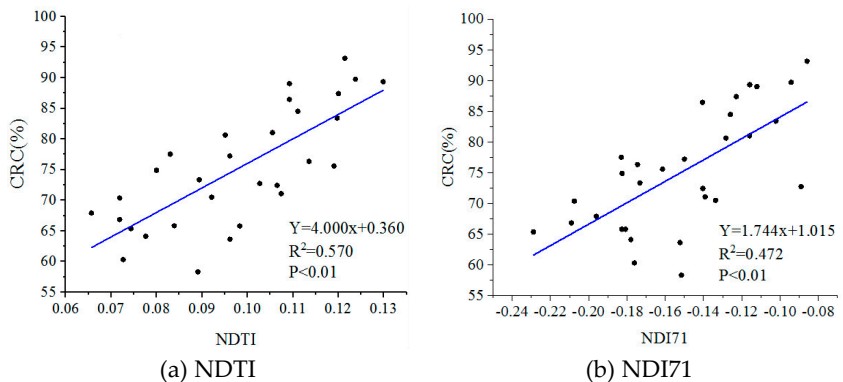

(a) NDTI　　　　　　　　　　　　　　　　(b) NDI71

**Figure 4.** Regression between CRC and some optical crop residue indices.

*3.2. Correlation between Radar Parameters and Field Measured CRC*

Six radar parameters were used to explore the ability of microwave data in CRC estimation. The regression equations between RPs and field measured CRC are shown in Table 3. Considering the $R^2$ values, the best and worst RPs were RI2 ($R^2$ = 0.430) and VV_ME ($R^2$ = 0.123), respectively. The other RPs had similar $R^2$ values between 0.30 and 0.40, with an order from highest to lowest of RI1, VH_ME, $\sigma^0_{VV}$ and $\sigma^0_{VH}$. The scatterplots between CRC and some radar parameters are shown in Figure 5.

**Table 3.** The regression relationships between CRC and radar parameters.

| Radar Parameters | Regression Equations | $R^2$ | RMSE (%) |
|---|---|---|---|
| $\sigma^0_{VV}$ | Y = −1.576x + 0.903 | 0.341 ** | 8.086 |
| $\sigma^0_{VH}$ | Y = 9.177x + 0.638 | 0.319 ** | 8.241 |
| RI1 | Y = −0.010x + 0.848 | 0.365 ** | 8.515 |
| RI2 | Y = −0.117x + 0.753 | 0.430 ** | 7.052 |
| VV_ME | Y = −0.001x + 0.875 | 0.123 n.s. | 9.327 |
| VH_ME | Y = 0.001x + 0.628 | 0.359 ** | 7.955 |

**Note:** Probability levels are indicated by n.s. and ** for "not significant" and $p < 0.01$, respectively. $\sigma^0_{VV}$ and $\sigma^0_{VH}$ means the normalized backscattering coefficients in VV, and VH polarization directions. RI1 and RI2 means radar indices defined in formula (1) and formula (2). VV_ME and VH_ME means the mean variables of $\sigma^0_{VV}$ and $\sigma^0_{VH}$ in gray level co-occurrence matrix.

To validate the estimation accuracy of the regression equation, RMSE was used to compare the field measured CRC and predicted CRC (Table 3). Table 4 shows that RMSE values ranged from 7.052% to 9.327%. The order from lowest to highest was RI2, VH_ME, $\sigma^0_{VV}$, $\sigma^0_{VH}$ , RI1 and VV_ME, which had some difference from that of $R^2$ value. Nonetheless, RI2 turned out to be the most accurate RPs for CRC estimation.

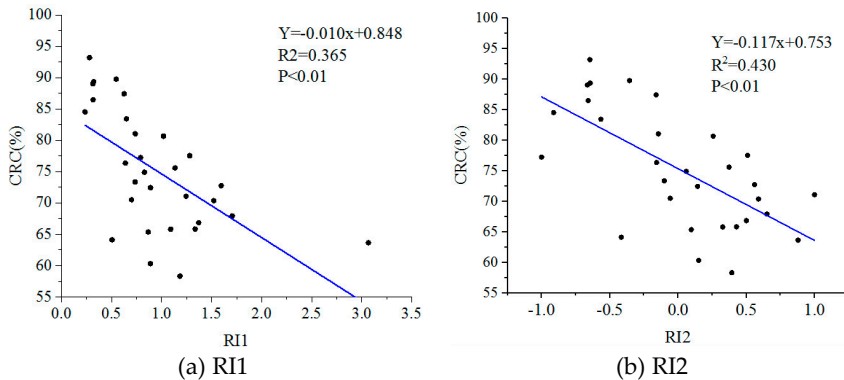

**Figure 5.** Regression between CRC and some radar parameters.

**Table 4.** The regression relationships between CRC and OCRI-RPs.

| Variables | Regression Equations | $R^2$ | RMSE (%) |
|---|---|---|---|
| NDRI $\times \sigma^0_{VV}$ | Y = 0.300x + 0.647 | 0.625 ** | 6.296 |
| NDI7 $\times \sigma^0_{VV}$ | Y = 0.303x + 0.662 | 0.552 ** | 6.846 |
| NDTI $\times \sigma^0_{VV}$ | Y = 0.298x + 0.643 | 0.671 ** | 5.741 |
| NDI71 $\times \sigma^0_{VV}$ | Y = 0.308x + 0.634 | 0.667 ** | 5.763 |
| NDRI $\times \sigma^0_{VH}$ | Y = 0.366x + 0.685 | 0.567 ** | 6.528 |
| NDI7 $\times \sigma^0_{VH}$ | Y = 0.451x + 0.685 | 0.576 ** | 6.599 |
| NDTI $\times \sigma^0_{VH}$ | Y = 0.324x + 0.690 | 0.585 ** | 6.435 |
| NDI71 $\times \sigma^0_{VH}$ | Y = 298x + 0.691 | 0.507 ** | 7.097 |
| NDRI $\times$ RI1 | Y = 0.306x + 0.633 | 0.672 ** | 5.789 |
| NDI7 $\times$ RI1 | Y = 0.309x + 0.651 | 0.595 ** | 6.423 |
| NDTI $\times$ RI1 | Y = 0.294x + 0.634 | 0.728 ** | 5.221 |
| NDI71 $\times$ RI1 | Y = 0.300x + 0.624 | 0.683 ** | 5.578 |
| NDRI $\times$ RI2 | Y = 0.419x + 0.631 | 0.690 ** | 5.630 |
| NDI7 $\times$ RI2 | Y = 0.433x + 0.647 | 0.623 ** | 6.198 |
| NDTI $\times$ RI2 | Y = 0.393x + 0.635 | 0.738 ** | 5.140 |
| NDI71 $\times$ RI2 | Y = 0.403x + 0.625 | 0.696 ** | 5.473 |
| NDRI $\times$ VH_ME | Y = 0.395x + 0.685 | 0.568 ** | 6.606 |
| NDI7 $\times$ VH_ME | Y = 0.443x + 0.689 | 0.525 ** | 6.936 |
| NDTI $\times$ VH_ME | Y = 0.334x + 0.692 | 0.551 ** | 6.755 |
| NDI71 $\times$ VH_ME | Y = 0.326x + 0.692 | 0.515 ** | 7.522 |

**Note**: OCRI-RPs means the indices combined optical crop residue indices and radar parameters; ** means model significant at the 0.01 probability level ($p < 0.01$).

### 3.3. Correlation of Combined Optical Crop Residue Indices and Radar Parameters with CRC

Based on the correlation between satellite derived variables and CRC, NDRI, NDI7, NDTI, NDI71, could predict CRC more accurately than other OCRIs in this study. Additionally, $\sigma^0_{VV}$, $\sigma^0_{VH}$, RI1, RI2 and VH_ME performed relatively better than other RPs. Thus, after normalizing these four OCRIs and five RPs, they were used to establish the combined optical crop residue indices and radar parameters (OCRI-RPs) by multiplying each selected OCRIs with each selected RPs. In total, it produced 20 OCRI-RPs. The regression models of OCRI-RPs for CRC estimation are shown in Table 4. It showed that NDTI $\times$ RI2 had the highest $R^2$ value of 0.738, which increased by 0.168 when compared with the best results in OCRIs and RPs ($R^2$ = 0.570). Meanwhile, two OCRI-RPs had $R^2$ values above 0.70, eight OCRI-RPs had $R^2$ values between 0.60 and 0.70, and ten OCRI-RPs had $R^2$ values below 0.60. In most cases, OCRI-RPs performed better than OCRIs or RPs in CRC estimation, with the exception of NDTI $\times$ VH_ME, which had a $R^2$ value ($R^2$ = 0.551) lower than that of NDTI ($R^2$ = 0.570). The scatterplots between CRC and some OCRI-RPs are shown in Figure 6.

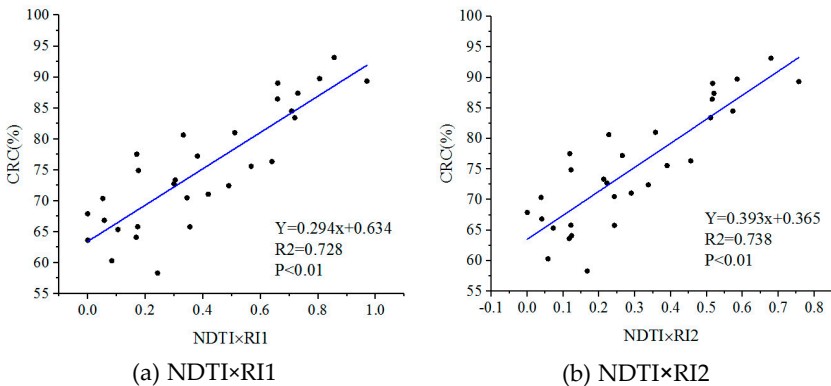

**Figure 6.** Regression between CRC and come combined OCRI-RPs.

RMSE of leave-one-out cross validation can represent the ability of regression models in predicting CRC beyond training set. The experimental results showed that RMSE values of OCRI-RP$_S$ were between 5.140% and 7.522% (Table 4). NDTI × RI2 and NDI71 × VH_ME had the lowest and highest RMSE value (RMSE = 5.140% and 7.522%, respectively). The decrease of RMSE values proved the effectiveness of OCRI-RPs in CRC estimation.

### 3.4. CRC Estimation Based on Optimal Subset Regression

Optimal subset regression is a commonly used multiple regression method, which considers all combinations of independent variables to estimate dependent variables. For the 20 OCRI-RPs, it can construct 1,048,575 alternative submodels. R software would present optimal submodels with one to eight independent variables. After normalizing and calculating the sum of four evaluation indices (AIC, BIC, $R^2$, RMSE) in one submodel, it can obtain the best submodels. The detailed information of the eight submodels are shown in Table 5. It could easily find that submodel with two independent variables had a highest composite index. Therefore, it was the optimal model for CRC estimation. The equation using model 2 to estimate CRC is shown in Equation (6):

$$CRC = 0.638 + 0.210 \times NDI71\_\sigma^0_{VV} + 0.175 \times NDTI\_\sigma^0_{VH} \tag{6}$$

where $NDI71\_\sigma^0_{VV}$ represents $NDI71 \times \sigma^0_{VV}$ and $NDTI\_\sigma^0_{VH}$ represents $NDTI \times \sigma^0_{VH}$. The variance inflation factor (VIF) of $NDI71 \times \sigma^0_{VV}$ and $NDTI\_\sigma^0_{VH}$ in optimal model were both 1.665, which proved optimal model did not have a colinearity problem. The $R^2$ value of the optimal model was 0.770 and the RMSE of LOOCV was 4.846%. They both had certain improvements when compared with the best results in Table 5 ($R^2$ = 0.738, RMSE = 5.140%). Thus, the optimal model would be used as the final model to estimate CRC in study area. Figure 7 shows the scatterplot between field measured CRC and optimal model estimated CRC.

**Table 5.** Alternative submodels and evaluation indexes of OCRI-RPs by optimal subset regression.

| Model | Adj $R^2$ | AIC | BIC | RMSE | Composite Index | Selected Independent Variables in Model |
|---|---|---|---|---|---|---|
| 1 | 0.73 | −3.04 | −33.34 | 5.14 | 2.34 | NDTI × RVI2 |
| 2 | 0.75 | −3.86 | −33.89 | 4.85 | 3.26 | NDI71 × $\sigma^0_{VV}$, NDTI × $\sigma^0_{VH}$ |
| 3 | 0.76 | −3.33 | −32.76 | 4.83 | 3.17 | NDI71 × $\sigma^0_{VV}$, NDTI × $\sigma^0_{VH}$, NDI71 × $\sigma^0_{VH}$ |
| 4 | 0.77 | −2.52 | −31.33 | 4.78 | 2.99 | NDI7 × RVI1, NDRI × $\sigma^0_{VV}$, NDI7 × $\sigma^0_{VH}$, NDTI × $\sigma^0_{VH}$ |
| 5 | 0.77 | −1.10 | −28.94 | 5.08 | 2.01 | NDRI × RVI1, NDI71 × RVI1, NDRI × $\sigma^0_{VV}$, NDI71 × $\sigma^0_{VV}$, NDI71 × $\sigma^0_{VH}$ |
| 6 | 0.77 | −0.05 | −27.28 | 5.42 | 1.17 | NDRI × VH_ME, NDRI × $\sigma^0_{VV}$, NDI71 × $\sigma^0_{VV}$, NDI71 × VH_ME, NDRI × RVI1, NDI71 × RVI1 |
| 7 | 0.79 | −0.17 | −28.15 | 5.73 | 1.31 | NDRI_RVI1, NDI71_$\sigma^0_{VV}$, NDRI_VH_ME, NDI71_$\sigma^0_{VH}$, NDRI_$\sigma^0_{VV}$, NDI7_VH_ME |
| 8 | 0.79 | 1.31 | −25.89 | 5.31 | 1.09 | NDRI_RVI1, NDRI_$\sigma^0_{VV}$, NDRI_$\sigma^0_{VH}$, NDI71_$\sigma^0_{VV}$, NDTI_$\sigma^0_{VH}$, NDI71_RVI1, NDTI_VH_ME, NDI71_VH_ME, |

**Note:** Adj $R^2$, AIC, BIC and RMSE represent adjusted coefficient of determination, Bayes Information Criterion, Akaike's Information Criterion and root mean square error of leave-one-out cross validation, respectively.

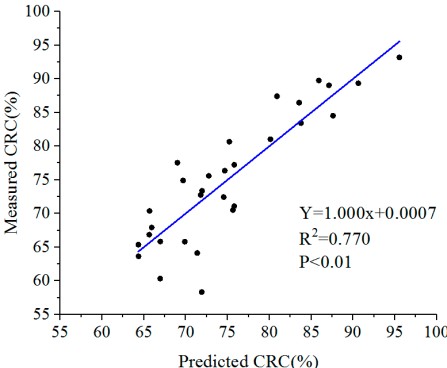

**Figure 7.** Scatterplot between field measured CRC and optimal model estimated CRC.

### 3.5. CRC Mapping in Study Area

Using two Sentinel-2 images, the winter wheat cultivation area could be extracted. Combining remote sensing image and cultivation area of winter wheat, the distribution of winter wheat residue coverage could be mapped by the optimal model based on $NDI71 \times \sigma_{VV}^0$ and $NDTI \times \sigma_{VH}^0$ indices (Figure 8). Statistical analysis has found that the average CRC was 68.05% and standard deviation of CRC was 6.36% for the study area on 14 June 2018. Generally, farmland with CRC less than 30% is defined as non-conservation tillage (intensive/conventional tillage), while farmland with CRC greater than 30% is considered conservation tillage [13]. It was a relief to find out that only 0.02% of the farmland was under non-conservation tillage in the study area. For the farmlands with CRC higher than 30%, they were generally divided into two classes: 30–60% and 60–100% [42]. The results showed that only 0.74% of the farmlands had a CRC of 30%–60% and 99.24% of farmlands had a CRC above 60%. Previous studies found roughly 65% of the soil erosion can be expected to control in the farmland when CRC is greater than 30%, while 90% of erosion can be avoided when CRC is higher than 60% [42,43]. This means that at least 90% of the surface soil erosion can be prevented for the high CRC areas in Yucheng County. This would make great contributions to the next year's crop cultivation. The results suggested that most of the crop residues were left on the farmland and the practices of conservation tillage had been promoted in Yucheng County.

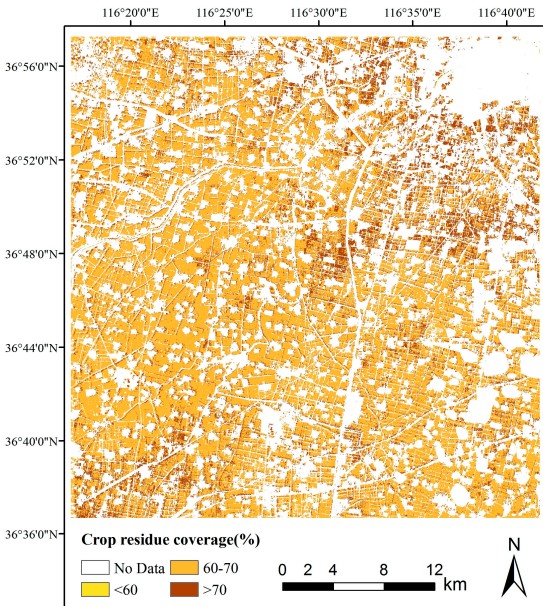

**Figure 8.** Predicted CRC map in study area on 14 June 2018.

## 4. Discussion

In this study, most of the optical crop residue indices used had certain correlations with field measured CRC, particularly NDTI (Table 3). NDTI had a highest correlation with field measured CRC ($R^2$ = 0.570 and RMSE = 6.560%). This result was coincidental with the study of Jin et al. in 2015 [37]. Previous studies have shown that crop residues and soil had different reflection characteristics in the 1450–1960 nm and 2000–2100 nm ranges [1,44]. The reflectance curve of winter wheat crop residues and soil collected by an Analytical Spectral Device (ASD) spectrometer in the laboratory (Figure 9) also proved these reflection characteristics for the reflectance of crop residues having a peek of 1650 nm, and a valley of 2100 nm, while that of soil were flat and had a peak. Daughtry et al. [1] indicted that water absorption in the 1450 nm and 1960 nm range made crop residues have a peak of reflectance in 1650 nm. The broad absorption feature of crop residues in 2100nm may be associated with cellulose and lignin. NDTI were calculated from the bands b11 (1540–1680 nm) and b12 (2080–2320 nm) of Sentinel-2, which were sensitive bands for crop residue identification. Thus, NDTI performed better than any other OCRIs. NDRI was calculated by the b4 (645–683 nm) and b12 of Sentinel-2. It had a certain ability to estimate CRC. Gelder et al. [19] proved that NDRI performed better than NDTI when green vegetables appeared in farmland. However, in this study, there were little green vegetables on the farmland. Thus, the use of b4 made it more difficult to estimate CRC. NDI7 was calculated by the b8 (763–907 nm) and b12 bands of Sentinel-2. It has been proven that near infrared bands were sensitive to plant structure [37,45], which gives NDI7 the potential to distinguish crop residues from soil. Among the four new OCRIs, NDI71 performed better than NDI7, while NDI72, NDI73 and NDI74 performed worse than NDI7. It suggested that narrow bands were more sensitive to the reflectance of winter wheat. Therefore, more attention should be paid to the use of narrow bands for CRC estimation.

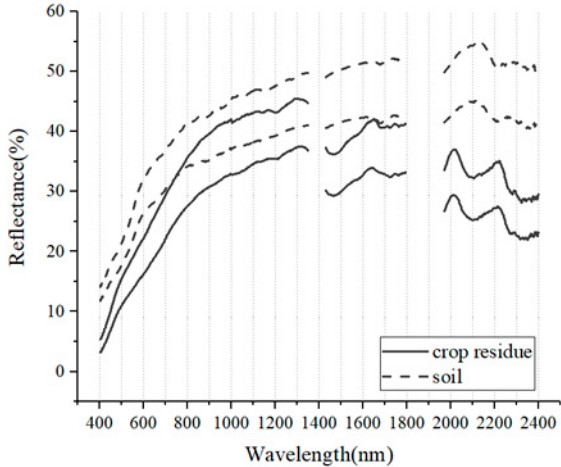

**Figure 9.** The spectral curve of winter wheat residues and soil.

This study has proven that backscattering coefficients in the VV polarization direction and VH polarization direction have certain correlations with field measured CRC ($R^2$ = 0.341, RMSE = 8.086% and $R^2$ = 0.319, RMSE = 8.241%). The results were similar to the findings of McNairn et al. in 1992, who held the view that cross-polarized backscattering coefficients can estimate CRC well and backscattering coefficients in the VV polarization direction can distinguish crop residues from soil [24]. However, the $R^2$ values were not very high between CRC and backscattering coefficient. It may because the incidence angles of Sentinel-1 images used in this study were both around 39°, but the best angle for CRC estimation was between 40° and 50° [26]. Radar indices have been popular indices for vegetable water content estimation and fresh weight estimation [46,47]. This study tentatively explored the potential of radar indices for CRC estimation, and interestingly found that RI1 and RI2 both performed better than $\sigma^0_{VV}$ and $\sigma^0_{VH}$, which proved the effectiveness of radar indices in CRC estimation. Sentinel-1 images data only have two polarization combinations. It has limited the study on

exploring the relationships between CRC and radar indices constructed by backscattering coefficients in the other two polarization direction, which needed to be further explored in other microwave data.

Based on the well performed OCRIs and RPs, there were twenty OCRI-RPs which have been developed to estimate CRC. Most of the OCRI-RPs had a relatively higher $R^2$ value and a relatively lower RMSE value when compared with the OCRIs and RPs, which suggested that it was effective to improve the accuracy of CRC estimation by combining optical information and microwave information. For the purpose of further improvement of model accuracy, this study used optimal subset regression to estimate CRC. As an alternative method of stepwise regression technology, optimal subset regression can well fit the experimental data. It uses all combinations of independent variables to fit dependent variables, which can solve the problem of inconsistency made by forward stepwise regression and backward stepwise regression. The final results showed that the combinations of $NDI71 \times \sigma_{VV}^0$ and $NDTI \times \sigma_{VH}^0$ had the highest correlations with field measured CRC ($R^2$ = 0.770, RMSE = 4.846%). It had significant improvements in CRC estimation, when compared with the best results with OCRIs ($R^2$ = 0.570 and RMSE = 6.560%), the best results with RPs ($R^2$ = 0.430 and RMSE = 7.052%) and the best results with OCRI-RPs ($R^2$ = 0.738, RMSE = 5.140%). It was seen that synergistic use of optical and microwave data is very helpful to improve the accuracy of CRC estimation, which can open a new way for CRC estimation.

As the result shows in Figure 8, CRC in Yucheng County has been divided into three classes: 0–60%, 60–70%, and 70–100%. It could be easily found that most of the farmland in Yucheng County had a CRC between 60% and 70%. In order to further explore the relationship between the results and various satellite derived variables, this study calculated the mean values of some satellite derived variables in three classes (Table 6). The results showed that the mean values of NDTI, $NDI71 \times \sigma_{VV}^0$, $NDTI \times \sigma_{VH}^0$ were positively correlated with CRC. The results of NDTI were coincidental with the results in Figure 4. It proved that NDTI, $NDI71 \times \sigma_{VV}^0$, $NDTI \times \sigma_{VH}^0$ were good variables to estimate CRC. The mean value of NDI71 between 60% and 70% and the mean value of NDI71 between 70% and 100% were similar to results in Figure 4. However, it did not have an appropriate mean value between 0% and 60%, which means it performed poorly in the estimation of low coverage areas. The mean values of RI1 and RI2 did not have a good relationship with CRC. They also performed poorly in low coverage areas. In conclusion, NDTI was the best variables in CRC estimation when just using spectral information or SAR information.

**Table 6.** The mean value of some satellite derived variables in three classes.

| Satellite Derived Variables | 0–60% | 60–70% | 70–100% |
|:---:|:---:|:---:|:---:|
| NDTI | 0.03 | 0.08 | 0.12 |
| NDI71 | −0.18 | −0.17 | −0.15 |
| RI1 | 0.37 | 0.60 | 0.44 |
| RI2 | 0.19 | 0.19 | −0.13 |
| $NDI71 \times \sigma_{VV}^0$ | 0.02 | 0.05 | 0.30 |
| $NDTI \times \sigma_{VH}^0$ | −0.52 | 0.04 | 0.43 |

## 5. Conclusions

For the purpose of mapping crop residue coverage in Yucheng County, this study explored the potential of seven optical crop residues indices and six radar parameters in crop residue estimation. And then, the combined indices, which were calculated by multiplying each optical crop residues indices with each radar parameter, were proposed to estimate crop residue coverage. The optimal estimation model was built up by the combined indices with the optimal subset regression method and the crop residue coverage in Yucheng County was obtained by an optimal model. The results showed that NDTI was the best index for crop residue coverage estimation of the optical crop residue indices ($R^2$ = 0.570, RMSE = 6.560%), while RI2 performed best in radar parameters ($R^2$ = 0.430 and RMSE = 7.052%). Combined optical information with microwave information can significantly

improve the correlation between indices and field measured CRC, for most of the combined indices had improvements in prediction accuracy when compared with corresponding optical crop residue indices and radar parameters. Among the combined indices, NDTI × RI2 had a relatively higher $R^2$ value of 0.738 and a relatively lower RMSE value of 5.140%. The optimal multiple regression model was made up of NDI71 × $\sigma^0_{VV}$ and NDTI × $\sigma^0_{VH}$, which has the highest $R^2$ value of 0.770 and lowest RMSE value of 4.846%. The mean crop residues coverage in Yucheng County was 68.05%, which suggested conservation tillage had a great promotion in Yucheng County. The findings in this study demonstrated that synergistic use of optical remote sensing and microwave remote sensing is a good way to estimate CRC at a high accuracy. However, more studies are needed to test our method and findings in other regions for longer periods to achieve a general conclusion.

**Author Contributions:** The individual contributions made by every authors were shown below: conceptualization, W.C. and S.Z.; methodology and validation, W.C.; investigation and resources, W.C., Y.W. and F.P.; writing—original draft preparation, W.C.; writing—review and editing, W.C., S.Z., J.H. and Z.D.; supervision, S.Z., J.H. and Z.D.; project administration and funding acquisition, S.Z.

**Funding:** This research was funded by the Natural Science Foundation of China, grant number 41671429, the National Key R&D Program of China, grant number 2016YFB0502503 and the ISEF program of KFAS.

**Acknowledgments:** The authors thank Miss Dianmin Cong, Miss Xian Li, Miss Xiaoyu Liu, Mr. Zhaohua Zhang and Mr. Liang Shan for help in the field measurement work. The authors also thank the European Space Agency for providing Sentinel-1 and Sentinel-2 data free of charge.

**Conflicts of Interest:** The authors declare no conflict of interest.

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
