# Peer review of "Estimation of Winter Wheat Residue Coverage Using Optical and SAR Remote Sensing Images"

_remotesensing, doi:10.3390/rs11101163_

Round 1

Reviewer 1 Report

This study focused to estimate winter wheat residue coverage through Sentinel optical and SAR data. This research falls well within the scope of this journal. However, I have some concerns about the method, result presentations. I think it needs some major modifications and further reviews before it can be accepted for publication in this journal.

General comments:

1.      The author mentioned the 4 principles for selecting the best subset: coefficient of determination (R2),Bayes Information Criterion (BIC), Akaike's Information Criterion (AIC), model validation accuracy in section 2.4.2. It is necessary to give the four evaluation indices values for the best submodel in section 3.4.   

2.      This study explored the potential of seven optical crop residues indices and six radar parameters in crop residue estimation and the combined indices were proposed to estimate crop residues coverage. The optimal estimation model was built up and the crop residues coverage in Yucheng County was obtained by the model. However, I think the independent validation for the predicted CRC map (figure 8) should be carried out through more ground  investigation data.

3.      A color segmentation legend for the predicted CRC map may be a more direct and easy way of showing the study results than the present gradient color bar in figure 8. More details statistical analysis for the optical crop residue indices and radar parameters in different color segmentation of  CRC map should be added in section discussion, and it will helpful to explain your study results.  

Author Response

Thank you for the valuable comments. Our responses to reviewers’ comments are written in red color.

Reviewer Comments:

1. The author mentioned the 4 principles for selecting the best subset: coefficient of determination (R2),Bayes Information Criterion (BIC), Akaike's Information Criterion (AIC), model validation accuracy in section 2.4.2. It is necessary to give the four evaluation indices values for the best submodel in section 3.4.

Reply:Thank you for your valuable comments. The four evaluation indices values of the best submodel have been provided in Table 5 in page 10. 

2. This study explored the potential of seven optical crop residues indices and six radar parameters in crop residue estimation and the combined indices were proposed to estimate crop residues coverage. The optimal estimation model was built up and the crop residues coverage in Yucheng County was obtained by the model. However, I think the independent validation for the predicted CRC map (figure 8) should be carried out through more ground investigation data.

Reply: Thank you for this suggestion. For the field measured work was difficult, we did not have too much ground data. Considering that leave-one-out cross valiadation can make the best use of the ground data, we used this method to validate the predicted CRC. We think it may be a appropriate validation method when the number of samples were small. 

3. A color segmentation legend for the predicted CRC map may be a more direct and easy way of showing the study results than the present gradient color bar in figure 8. More details statistical analysis for the optical crop residue indices and radar parameters in different color segmentation of  CRC map should be added in section discussion, and it will helpful to explain your study results. 

Reply: Thank you for this suggestion. The Figure 8 has been modified in page 11 and the discussion about this content has been added in line 397-410.

Thanks again for your valuable suggestions. 

Reviewer 2 Report

PAGE 3, LINE 134-143 Field measurement data is confuse. You need improve this paragraph.

Page 5, line 165-168, you say: “which were acquired on June 13, 2017 and June 14, 2018 respectively”: improve this sentence.

Page 5 line 181: you say:”… And each of them has 13 spectral bands…” description of Sentinel-2 is known, public domain and does not require a description.  It is also suggested to delete table 1.

Table 5 and fig. 6 is confuse, why only simple regression (linear)

Fig 8, the map does not have geographic coordinates.

Page 11, line 336-337, you say: “… In this study, most of the optical crop residue indices used had certain correlations with field…” Why only Optical crops residue? What happen with microwave crop residue?

Author Response

Thank you for the valuable comments. Our responses to reviewers’ comments are written in red color.

1. PAGE 3, LINE 134-143 Field measurement data is confuse. You need improve this paragraph.

Reply: Thanks for your suggestion. The paragraph about the field measurement has been improved in line 135-148.

2. Page 5, line 165-168, you say: “which were acquired on June 13, 2017 and June 14, 2018 respectively”: improve this sentence.

Reply: Thanks for your suggestion. This sentence has been improved in line 168-169.

3. Page 5 line 181: you say:”… And each of them has 13 spectral bands…” description of Sentinel-2 is known, public domain and does not require a description.  It is also suggested to delete table 1.

Reply: Thanks for your suggestion. Some description of Sentinel-2 and the table 1 has been deleted.

4. Table 5 and fig. 6 is confuse, why only simple regression (linear)

Reply:Thanks for your suggestion. Because the relationship between field-measured CRC and the combined optical crop residue indices and radar parameters (OCRI-RPs) can be well expressed by simple linear models. Using exponential models or logarithm models did not have a higher R2 or a lower RMSE. Therefore, this study just used simple linear models to estimate CRC by OCRI-RPs.

5. Fig 8, the map does not have geographic coordinates.

Reply: Thanks for your suggestion. The geographic coordinates has been added in Figure 8.

6. Page 11, line 336-337, you say: “… In this study, most of the optical crop residue indices used had certain correlations with field…” Why only Optical crops residue? What happen with microwave crop residue?

Reply: Thanks for your suggestion. Because I want to talk about these two kinds of indices in two paragraphs, I did not mention radar parameters in this paragraph. The discussion about radar parameters was shown in line 367-381.

Thanks again for your valuable suggestions. I am looking forward to receiving your decision.

Reviewer 3 Report

This paper describes a method to combine remote sensing information from multiple sensors using statistical techniques to determine the amount of crop residue coverage in northern China.  The main concern with the paper is the style of the English writing which needs to be improved substantially to make the paper more readable.  Examples where there needs to be corrections include:

Line 124: locates should be located

Line 132-133: awkward sentence

Line 135: Need a better explanation of "photo method"

Line 140: samples should be sample

Line 147: Firstly should be "First"

Line 152: unclear "to extract soil from photo" should be "to identify soil pixels from the photograph?" 

Line 168: sentences should not begin with "And"

Line 227: Equation (4) seems out of place should be after line 226

There are many more examples where the English style needs improvement so I recommend extensive checking of English tense, number (singular vs plural), etc. before publication. I hope these revisions will make the paper easier to understand.

Author Response

Thank you for the valuable comments. Our responses to reviewers’ comments are written in red color.

1. Line 124: locates should be located

Reply: Thanks for your suggestion. This word has been modified in line 125.

2. Line 132-133: awkward sentence

Reply: Thanks for your suggestion. This sentence has been deleted.

3. Line 135: Need a better explanation of "photo method"

Reply: Thanks for your suggestion.More explanation about "photo method" has been added in line 143-146.

4. Line 140: samples should be sample

Reply: Thanks for your suggestion. This word has been modified.

5. Line 147: Firstly should be "First"

Reply: Thanks for your suggestion. This word has been modified in line 152.

6. Line 152: unclear "to extract soil from photo" should be "to identify soil pixels from the photograph?"

Reply: Thanks for your suggestion. This sentence has been improved in line 156-157.

7. Line 168: sentences should not begin with "And"

Reply: Thanks for your suggestion. This sentence has been improved in line 170.

8. Line 227: Equation (4) seems out of place should be after line 226

Reply: Thanks for your suggestion. The equation(4) has been put after line 226.

9. There are many more examples where the English style needs improvement so I recommend extensive checking of English tense, number (singular vs plural), etc. before publication. I hope these revisions will make the paper easier to understand.

Reply: Thanks for your suggestion.This paper has been checked again and hope that it has become more easier to understand.

Thanks again for your valuable suggestions. We are looking forward to receive your decision.

Round 2

Reviewer 1 Report

The manuscript has been significantly improved by the authors. The quality of presentation, as well as the scientific soundness have been increased significantly.  I think the manuscript will be ready for publication.

This manuscript is a resubmission of an earlier submission. The following is a list of the peer review reports and author responses from that submission.